# The Role of Housing in Facilitating Middle-Class Family Practices in China: A Case Study of Tianjin

**Lu Wang \* and Rose Gilroy**

School of Architecture, Planning & Landscape, Newcastle University, Newcastle upon Tyne NE1 7RU, UK;
r.c.gilroy@newcastle.ac.uk
\*  Correspondence: b5053772@ncl.ac.uk or lulullw2021@outlook.com

**Abstract:** This paper, drawn from a wider doctoral study that investigates how middle-class Chinese families manage and balance their resources to negotiate family duties across generations, focuses on the role of home ownership and property. The research considers intergenerational equity, which is a key part of social sustainability, and uses this to explore the shifting care expectations between generations and the inherent tensions between socioeconomic opportunities that have changed the shape of families and the belief in the importance of the family unit as a vehicle to deliver care. The research draws on the narratives of whole families in a ten-family study undertaken in the Chinese city of Tianjin. The findings reveal the critical role of housing resources in presenting alternative solutions to the performance of care. Firstly, the opportunity to make new choices in the face of shifting priorities across the life course is facilitated by property ownership. Secondly, it facilitates the possibility of living close by, but not together, maintaining the privacy of the nuclear family, but fulfilling care roles. Thirdly, housing resources promote variations on the traditional co-residence pattern for supporting frail elders and, finally, new forms of co-residences where care flows to the young family and their children.

**Keywords:** property ownership; family practices; Chinese middle class; co-residence; eldercare

## 1. Introduction

In the shadow of COP 26, the common parlance of sustainability is dominated by ecological concerns. World leaders and city fathers may also be focused on prosperity, understood as economic sustainability. The third pillar of social sustainability is more frequently overlooked [1]. Concerned as it is with well-being and social justice, it is, however, an apt lens through which to explore changing family practices in urban China, where rapid socioeconomic shifts have delivered employment mobility, higher education opportunities, and wealth accumulation chances never before experienced [2]. These changes are occurring in the context of a rapidly ageing population which, according to the United Nations, will account for almost 20% of all older adults on the planet, and 25% of all Chinese [3]. The East Asian model of family care, based on filial piety, is challenged by the shrinking human resources caused by the one-child policy and family dispersal. Care alternatives, such as residential-based care, are targeted at the very wealthy [4–6]. In an era of single children and greater employment mobility, the family unit may seek support from home and community-based services, but the quality, particularly of the former, is uneven and badly needs to be professionalised. Fu and Chui [7] discovered that living arrangements were linked to the requirements for home and community-based care services. The study found that the nearer the adult children lived to their older parents, the fewer the number of times the seniors requested community services, suggesting that the preferences of older people may lie with family-based care. Lau and Kirby [8] discuss the link between living arrangements and preventative eldercare. Chen and Chen [9] studied the living arrangement preferences of seniors in Taiwan, taking into account three factors: the older person's health, family resources, and social participation.

In this study, the concept of family resources draws on Bourdieu's theory of capitals. Economic capital, in terms of income, wealth, and intergenerational transfers, is important in determining life trajectories [10] (pp. 78–94). In this study, this is seen in salaries and property ownership. Social capital, which embraces knowledge, know-how, and connections, can also be mobilized to open up opportunities or gain advantages. In this study, it was apparent that middle-class well-educated professionals used their know-how to access medical and care resources to fit their needs. Finally, cultural capital, often defined as taste and cultural disposition, also includes ways of being and doing that embrace the ways of performing family. In this study, cultural capital is seen in how families negotiate the expectations of older generations, their presentation of self to their peers as filial children, and their understanding of their own potential care choices. The study employed a case-study approach in the Tier 2 city of Tianjin. The empirical data were taken from semistructured in-depth interviews with ten families, carried out with the adult members of three generations: grandparents, parents, and adult children. Taking a grounded theory approach, the research explored the patterns of family support and the generational differences between the expectations of care and the resource bases that could be deployed. This paper offers insights into the salience of housing, not simply as a wealth accumulator that can be transferred across generations, but also as an enabler of preferred family practices in the face of less attractive or unaffordable alternatives.

## 2. Housing Resources and Mismatched Housing Needs

In today's China, almost 90% of households own their own home [11]. At the same time, more than 20% of Chinese households own several properties, which is higher than in many developed countries [12]. Only fifty years ago, China was recognized for its communist welfare-oriented housing system, which distributed public rental homes to urban residents centred around their work unit, which was the mechanism through which citizens accessed not only housing, but also education, care for preschool children, and eldercare, thus giving women, in particular, the opportunity to engage more fully in waged work [13]. In 1998, the government proclaimed the end of state involvement in allocating housing [14] and, in the short period since, China has shifted from a society dominated by public renters to one of the countries with the greatest rate of homeownership [13].

### 2.1. The Rise of Inequality Driven by Housing

The political step to privatize public housing started in the UK in 1980 [15] (p. 31) and was adopted by many former Soviet bloc countries that were transitioning from Soviet and state control to more outward-facing democracies [16] (pp. 71–73). Later in China, homeownership growth was brought about by selling existing public housing, both work units and then municipal housing stock to sitting tenants. It may be stated that the sale of work-unit dwellings offered limited choice. However, for those high-grade workers whose accommodation was well-located in central neighborhoods, there was a considerable advantage [17]. While the market valuations were not cheap, they were desirable properties in high-demand areas that had retained their market advantage [18–20]. More recently, there has been enormous growth in home ownership, fuelled by a booming real estate market. In 2019, total sales of the real estate market reached almost CNY 16 trillion and accounted for nearly 10% of China's GDP [21].

In a study by Wu et al. [17] (p. 448), it is argued that "both socio-economic characteristics and socialist institutions contribute to current housing inequality". State mechanisms, such as the hukou system, advantage those who have urban residency rights and militate against migrants and newcomers. Members of the Chinese Communist Party were more likely to be rewarded with larger housing units during the era of the state dominance of housing and were, therefore, economically advantaged in an era of privatization. Wu et al. [17] also hypothesize that human capital, understood as educational attainment, was also likely to be rewarded with higher-grade housing allocations, which translates into windfall wealth accumulation now and, in the modern China, possibly better paid

employment. Huang et al. [19] reveal that many cities now offer housing subsidies to the educated and talented, which may reduce the purchase price of new housing by 50%, as well as the benefit of a local hukou.

*2.2. The Mismatch between Housing Need and Supply*

There is now a higher rate of multiple-property ownership in China among middle-income urban dwellers than in other developed nations [20]. According to research by Huang et al. [20], among owners of multiple homes, the dominant reasons for purchase were to house family members (40%); investment (30%), and for use as holiday homes (16%). Thus, it has been shown that purchasing a property can be a financial investment, can facilitate new lifestyle opportunities, or can increase the life quality of other family members.

As seen in other countries, such as the UK, households are increasingly pursuing residential property as an investment strategy and, thus, housing has become a means to securing and acquiring capital, rather than a place to live in dignity, raise a family, and prosper within a community. This is because it is valued as a commodity rather than personal habitation [22]; in short, there has been a shift from use-value to exchange-value. A more professional and financialized investor mindset is forming, even when compared to the 2000s [23]. Housing property ownership in China has become, as it has in many Western countries, a sort of private insurance against numerous life-course hazards, such as unemployment or poverty in old age, in addition to being part of a significant accumulation of wealth [24,25]. While housing has played a unique role in mediating the movement of families, it also sits at the foundation of intergenerational inequality [24,25].

The research undertaken by Huang et al. [12] argues that, now, a "Rent-Generation" is mirroring trends in the UK and elsewhere, and "millennials", unable to finance homeownership, are postponing leaving home and forming partnerships [26,27]. Notably, homeowners are also renting places to live. This is the outcome of a mismatch in housing demands in terms of space, time, and function [12]. For example, young adults, who work in Tier 1 cities, such as Shanghai, may be homeowners in their hometown because their parents bought them an apartment on the occasion of their marriage. The older parents leave their home to look after their grandchildren, and their adult children rent another home for them in the megacity.

**3. The Role of Housing in Later Life and Middle-Class Family**

The house plays a nonreplacement role in a family. According to Clark et al. [28], house prices affect fertility in China. Their study analyses data from the China Household Finance Survey from 2013 to 2017 to show that a 1% rise in housing costs results in a 0.94-percentage-point drop in the probability of having a child under the age of two years old [28]. This means that accelerating house prices depresses population growth. In an era of increasing house prices, this suggests that the new freedom to have a second child will not necessarily lead to population growth, except perhaps for those with depths of wealth and who are eligible for intergenerational wealth transfers.

*3.1. Implications of Housing Location for Later Life*

In the context of the global phenomenon of ageing, there is a growing understanding of the role of the home, and much research has demonstrated the special relationship that older people may have with their homes as mobility declines and they become more reliant on the home as a physical arena and social centre [29] (p. 79). While new marketized forms of housing, many of which have varying levels of care, are appearing in many countries, including in China, the majority of older people seem to prefer to age in their existing home [30]. Beyond the high use-value attached to the home by older people, as in other marketized housing countries, those with high exchange values have increasing options to use these resources to create the later life choices they prefer.

*3.2. The Role of Housing in Maintaining Middle-Class Advantage*

Gentrification is the phenomenon of neighbourhood improvement, frequently conducted through the displacement of lower-income groups by those with greater economic resources. Bridge [31] points out that gentrification typifies the techniques of social and cultural reproduction through the performance of a middle-class habitus in redeveloped neighbourhoods of the inner city, building on theories of differentiation. As a result, these gentrified neighbourhoods demarcate and replicate typical urban patterns that support structuralized norms and resources [32]. They can be seen as ways of cultural replication that often lead to social reproduction by sanctifying these cultures as sites of middle-class remaking that reproduce social spaces [32]. That is to say, "home-making, design, cultural and consumption-oriented practices reproduce both place and class" [32] (p. 3).

In China, this gentrification is driven by school districts. Education has always been an important factor in assigning housing, and other economic rewards, even in the former state-dominated housing system [17]. The free market produces a meritocracy in which highly educated people are rewarded with more resources [17]. Separating the impact of the market from the effect on housing inequality is difficult [17], but Wu et al. [33] investigate a procedure known as *jiaoyufication*, which entails buying (often at a high cost) an apartment in the catchment area of a prestigious elementary school. *Jiaoyufication*, "an extension of gentrification"—urban transformation fuelled by a demand for high-quality education—is displacing past lower-class residents, while simultaneously transforming old blue-collar neighborhoods into modern middle-class communities [32]. *Jiaoyufiers* are more interested in gaining cultural capital and transmitting it to their children, thus maintaining their class position [31]. Their investment in housing is driven by the need to give their children or grandchildren advantages that will lead to a prosperous future for their children and themselves.

## 4. Site Selection

Fieldwork (from January to July 2019) was carried out in the Tier 2 city of Tianjin. It was chosen for its under-researched quality, its concentration of middle-class people, and its high ageing population that ranks the city at No. 3 in China [34]. By the end of 2018, the Tianjin residential population aged 60 years and over had reached 2.46 million, accounting for 23.4% of the total residential population, an ageing rate far more than the national average, and third to Beijing and Shanghai [35]. The first author's own biographic connection to Tianjin provided an eased entry into the research.

## 5. Recruiting Participants

The need to understand care from the vantage point of different generations meant that the participant families needed to have at least two generations of adults. The desire to explore family practices within the emergent middle class meant that participants also needed to have at least one generation that might be identified as middle-class. Finally, these families needed to be willing to share their family stories with the researcher. In the context of China, this type of qualitative research is less usual, and asking entry into a family can easily be seen as intrusive.

Class identity in China is growing in complexity. An urban hukou for Tianjin is a given, but, in addition, class identity is still very much determined by income and, as such, is different for different tier cities, reflecting both local earnings and the cost of living. In Tianjin, according to the Hurun Report [36], those whose annual incomes range between CNY 100–500 K (about GBP 11,401 to 57,007) would be counted as middle-class. Talking about income can be difficult in many cultures so, in this study, participants were asked to identify a range that their annual salary fell within: more than CNY 100 k CNY but less than CNY 200 k.

Though income seems to dominate, there is an increasing construct of middle-class identity that includes Weberian and Bordieuan perspectives (see Table 1).

**Table 1.** The criteria of the recruited sample.

| Criterion | Requirements |
| --- | --- |
| Annual income | Minimum CNY 100 K per year (about GBP 11,401). |
| Occupation | Specialist or skilled job, or work at a managerial, or comparable, level, either in the state or private sectors. |
| Education | Above bachelor's degree or college degree. |
| Hukou | Urban household registration |
| House/car | Own a house and/or car, either with a mortgage or outright ownership. Lives in a gated community. |
| Self-identity | Seeks a lifestyle rather than a living and interested in self-exploration and self-actualisation. |

## 6. Methods

This paper draws on a broader ten-family (family details, see Table 2) investigation to explore how new family practices, particularly with respect to older people, are emerging in middle-class Chinese families. A case study methodology was chosen to understand meanings and actions, and how people construct them via the interpretation of studied everyday life rather than any prior hypothesis [37] (p. 17). This paper provides insights into the housing arrangements in the life courses of Chinese urban dwellers that are in line with family practices on eldercare, grandchildren care, and mutual support. Semistructured interviews were held in the participants' homes in order to increase their feelings of control, and for gaining insights into how the homes were organized [38], providing a route for participants to tell their stories, with their narratives being analysed in order to understand their behavior and meaning-making [38]. Multiple interviews took place in each family.

**Table 2.** Interviewed families and their properties.

| Family | Generations and Ages | Incomes of Households (Per Year) and Property Ownership |
| --- | --- | --- |
| Li | Mr. G1—86, Mrs. G1—85. | G1 around CNY 100 K (about GBP 11 K). G1 own one apartment for their occupation. |
| | Mr. G2—61, Mrs. G2 (D1)—62. | G2 around CNY 250 K (about GBP 27 K). G2 own two apartments, one for their occupation and one for their daughter (G3-D). |
| | G3-wife (34, daughter), husband (35). G4—two girls. | G3: CNY 300 K (about GBP 33 K) |
| Hao | Mr. G1—82. | G1 around CNY 80 K (about GBP 8 K). G1 owns one apartment for occupation. |
| | Mr. G2—62, Mrs. G2 (D1)—60. | G2 around CNY 400 K (about GBP 43 K). G2 own three apartments, one for living, one for a holiday home, and one in a good school district for their granddaughter's, G4's, education. |
| | G3-wife (34, daughter), husband (34). G4—one girl was of primary-school age. | G3: CNY 650 K (about GBP 71 K) and own one apartment for G3/G4's occupation. |
| Xing | Mrs. G1—83. | G1 around CNY 60 K (about GBP 6 K). G1 owns two apartments; one for Mrs. G1's occupation, and another for G2-S1's occupation. |
| | Mr. G2(S1)—62, Mrs. G2—60. | G2 around CNY 80 K (about GBP 8 K). G2 live in one of G1's apartment. |
| | G3-wife (34, daughter), husband (36). | G3: CNY 150 K (about GBP 16 K). G3 live with husband's mother, who owns the flat. |
| Huo | Mr. G1—79, Mrs.G1—80. | G1 around CNY 70 K (about GBP 7 K). G1 live with their second daughter (G2-D2), and the apartment was bought by their youngest son (G2-S3). |
| | Mr. G2 (S1)—56, Mrs. G2—60. | G2 around CNY 220 K (about GBP 24 K). G2 own two apartments, one for living, one for their son (G3-S). |
| | G3-S (25, son). | G3: CNY 130 K (about GBP 14 K). |

**Table 2.** *Cont.*

| Family | Generations and Ages | Incomes of Households (Per Year) and Property Ownership |
|---|---|---|
| Ye | Mr. G1—85, Mrs. G1—84. | G1 around CNY 100 K (about GBP 11 K). G1 own one apartment for living. |
| | Mr. G2—60, Mrs. G2(D1)—60. | G2 around CNY 180 K (about GBP 19 K). G2 own two apartments, one for living, and one is rented to increase income. |
| | G3-wife (33), husband (33, son). | G3: USD 80 K (about GBP 61 K). G3 own a house in the US and were supported financially by G2. |
| Zhao | Mrs.G1—86. | G1 around CNY 70 K (about GBP 7 K). G1 own one apartment for living. |
| | Mr. G2—63, Mrs.G2(D1)—62. | G2 around CNY 250 K (about GBP 27 K). G2 own two apartments, one for living, and one for their daughter (G3-D) and her husband. |
| | G3-wife (daughter), husband. | G3: CNY 250 K (about GBP 27 K). |
| Wang | Mr. G1—87, Mrs. G1—87. | G1 around CNY 80 K (about GBP 8 K). G1 own one apartment for living. |
| | Mr. G2 (S1)—63, Mrs. G2—61. | G2 around CNY 250 K (about GBP 27 K). G2 own two apartments, one for living, and one is rented to earn more income. |
| | G3-wife (33, daughter). | G3: CNY 200 K (about GBP 22 K). G3 owns a flat in Beijing, which was financially underpinned by G2. |
| Kong | Mr. G2—60, Mrs. G2—59. | G2 around CNY 80 K (about GBP 8 K). G2 sold their flat to help G3 to buy their first apartment and they now live with G3. |
| | G3-wife (35, daughter), husband (41). G4—one boy, and one girl. | G3: CNY 400 K (about GBP 44 K). G3 own two apartments, one for G2/G3/G4 to live as an extended family and another for rent. |
| Han | Mr. G2—63, Mrs. G2—62. | G2 around CNY 80 K (about GBP 8 K). G2 own a flat in their hometown. |
| | G3-wife (36), husband (36, son). G4—one boy | G3: CNY 700 K (about GBP 77 K). G3 own a flat and they were planning to buy a school-district apartment. |
| Fu | Mr. G2—68, Mrs. G2—67. | G2 around CNY 220 K (about GBP 24 K). G2 own two apartments, one in Tianjin and one in Qingdao. |
| | G3-wife (40), husband (41, son). G4—one girl (12). | G3: CNY 800 K (about GBP 88 K). G3 own one apartment. |

The lead author spoke to the oldest generation (typically now in their eighties and labelled here as "G1"); their child or children, usually in their late fifties or early sixties, labelled "G2"; and the only child in their thirties, referred to as "G3". No attempt was made to speak to their child or children (G4) who were very young. In notation, a daughter is represented as "D", and "S" represents a son. Their order of seniority is also denoted, so "G2-D2" in the Li family is the second daughter in the second generation of the Li family. The data transcripts were written in Chinese to preserve the meaning of the language used in the interviews. The first version of the transcripts was sent to the interviewees to approve [39]. The subsequent agreed-upon transcripts were then translated into English.

The qualitative analysis software, NVIVO 11, was used to store and support data analysis through assigning nodes. The analysis followed the criteria of grounded theory in that the lead author read the narratives line-by-line, underlining noteworthy phrases or points. This was done until all viable concepts and categories were found, resulting in open coding. The coding and nodes were then compared and sorted to see whether there were relationships.

## 7. Findings

In this housing-focused substudy, four themes are highlighted and summarized. First, the salience of housing in promoting location changes in line with shifting life priorities. G2-D2, in the Li family, shared her experiences of deciding on optimum locations:

*I used to live at the city's south end, close to my workplace. Then, in 2000, I purchased a new apartment where the top schools are concentrated because I was more concerned with my son's (G3) education at that time. I could not stop worrying about my parents'*

*(G1) health until I was close to retirement age, so I decided to buy another flat since it is handy for me to visit my parents. I also hope that my parents will come to live with me (G2-D2, 60 years old, in the Li family).*

In the Fu family, G2 downsized to suit their changed lifestyle, which focused more on supporting their son and his family:

*Although we moved from the big home to this 70 m² apartment, life is so simple. Since the year our daughter-in-law was pregnant, we often travelled between Tianjin and Qingdao. After retirement, we felt the big home was not only unnecessary and also tricky to manage. Now we are delighted to live in this lovely apartment, and the space is enough for our everyday life. (G2 couple, Mr G2-68 and Mrs 67, in the Fu family).*

The second theme is that more middle-class family members choose to live apart but close by, thus achieving both privacy and support. Generation 2 in the Li family (there were three daughters in G2) started the housing conversation about buying an apartment for G1 to live in that was close to the eldest sister, that would be easily visitable, and that would allow all of the sisters to support their parents This intention was not realized because the G1 couple were hesitant to move away from their own neighborhood, which was conveniently close to the market and Mrs. G1's sister. It is significant that G1 have agency because they too are homeowners, whose needs and choices are not wholly dependent on their children. In the Hao family (see Figure 1), Mr. G1 was a lecturer at the university. His children (G2), except G2-S2, all work in the university, including G2-D1 and her husband, G2-D3, and G2-S4 and his wife. G2-D1 and G2-S4 live about a 20-min walk, or 5 min by car, from G1's home. The primary school and kindergarten are opposite G2's community, so it is convenient for G2-S4 to collect his son (G4) and visit G1 on the way. G2-D1 also sends her granddaughter to the same kindergarten.

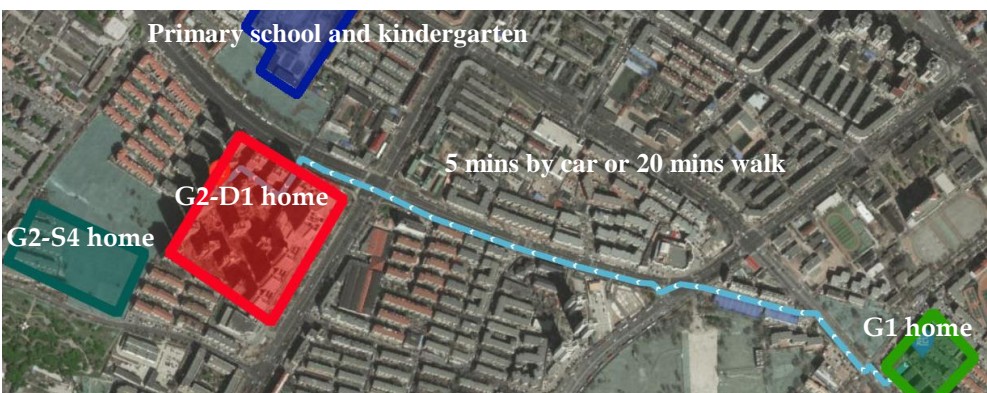

**Figure 1.** Locations of G1, G2-D1, and G2-S4 in the Hao family.

In the Xing family, Mrs. G1 has dementia and is seriously frail, needing 24-h care. Her son, G2-S1, lives in the same community as his mother and he fulfills the main caregiving while his siblings visit throughout the week.

In the Huo family, the G2 couple expected their son to live nearby and bought him an apartment. The new apartment is close to G2's home and G3's future workplace, following his anticipated graduation with a master's degree.

Third, there are new variations on the concept of "co-residence" to support frailer parents. An example is the daughters (G2) of the Zhao family, who move in with G1 on a rotation. G1 lives in Qiqihar city, located in the northeast of China. G2-D1 travels to her mother's home, which takes about 20 h by train (see Figure 2a below). The G3 couple (G2-D1's daughter and her husband) live close by their parents (see Figure 2b below).

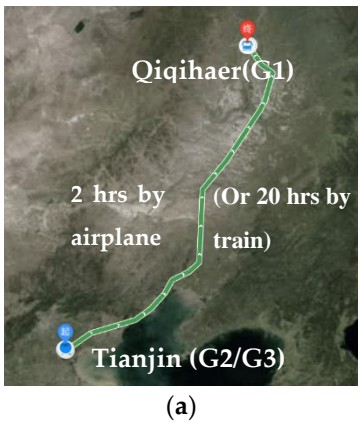

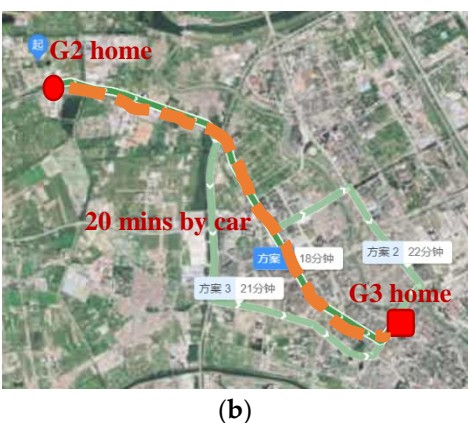

(**a**) (**b**)

**Figure 2.** This is a figure: (**a**) Time spent travelling from G2-D1′s home to G1′s home; (**b**) Time spent travelling from G2-D1′s home to their daughter′s home (G3).

In the Zhao family, Mrs. G1 was 86 years old in 2019 and was still in good health. She is a retired teacher, and she is always well-dressed. The three daughters of G2 have tried to take care of their mother in many ways. When G1 was over 80, her daughters decided to hire a helper to live with her. The idea had already been discussed over the five years or more since their father died. The issue then was that only G2-D3 lived in the same city as their mother, so it was challenging to meet their mother′s needs as they wanted. After the decision, their family practice was that G1 lived with a 24-h helper, and G2-D3 visited her regularly. However, three years ago, G2-D3 moved to a southwest city because of her business, and G2-D1 persuaded Mrs. G1 to live with her in Tianjin. Mrs. Zhao (G2-D1) shared that she, and her husband, had moved from her hometown to Tianjin more than 20 years ago, but felt guilty that she had avoided her duty to care for her parents and wanted to do something to repay them. She (G2-D1) assumed that her mother would like to live with them (G2/G3) because her home was also close to her younger sister′s house (G2-D2) in a town near Tianjin. Nonetheless, only a few months later, Mrs. G1 insisted on returning to her own home. Thus, the three sisters (G2) discussed hiring a care worker at home, and the three daughters, in turn, each came back to stay with Mrs. G1 for four months of the year:

> *I was used to living in my own home. It was good to live with my oldest daughter (G2-D1) because she has good hands for taking care of me, I know, but I prefer that she comes and visits me in my home. It is different . . . they (G2, G3) are so busy . . . they (G2) have to help [G3] to decorate the apartment, I can do nothing to help . . .* (Mrs G1, 86-year-old, in the Zhao family).

Independent elders may prefer living in their own place over living with an adult child, and this has been observed in the Zhao family. The assumption is that children should consider how they will care for older parents, but the evidence suggests that G1 has no wish to be passive but, rather, looks for ways to be useful.

The final theme is the extended family, with G2/G3/G4 co-residing to support the young people through childcare. The Kong and Han families, G2, were living with G3 and G4. They were expected to deliver support to their grandchildren. An expectation of active grandparenting is becoming normative in Tianjin, which further influences the attitudes of adult children towards their parents.

In the Kong family, Mrs. G3- complained that her parents had provided negligent care for her child. She regretted the decision to ask her parents to live with them:

> *My mother never looks after my son at all! My father wants to help me, but he is careless. I could not depend on them, so I quit my job and focused on taking care of him by myself. I do regret now that I asked my parents to live with me.* (G3-wife in the Kong family).

She (G3-wife) also revealed that, in asking her parents to sell their much smaller home and move to this three-room apartment, she expected them to be grateful and willing to help with childcare, which, in turn, would lead to her care of them in later years. Mrs. G2 accepted the proposition but refuses to shoulder the duty of grandchild care.

A more positive experience was voiced by Mrs. G3, in the Han family:

*My parents-in-law live with us to help to look after my son. With their help, I can focus on my work and have a competitive salary better than my husband's. I think they are satisfied living with us because they sleep in the best room in my home [she and her husband bought this flat], and my husband, my son and I sleep in the small bedroom, but I always appreciate their support and will repay this in their later life.*

These two contrasting examples demonstrate that the concept of filial piety, understood as the care duty owed to older parents, can no longer be seen as a given, but may now be seen as flowing two ways, with eldercare dependent on those G2 parents fulfilling their new care duties to an expected standard.

## 8. Discussion and Conclusions

This study has revealed that middle-class families have choices in care provision that are facilitated by home ownership, and that this care provision extends to Generation 1 and to Generations 3 and 4. A number of narratives highlight that the life-course changes and the emergent priorities, to care for older parents, to provide more support to a son and his wife, or to improve a grandchild's education chances, were facilitated by home ownership that provides choices. These considerations are present across the life course, and the resources for dealing with them are traced in everyday practices, e.g., opportunities for better employment, eldercare, children rearing, and education.

Education emerges as a significant determinant for middle-class families. The greater affluence of Generation 1 means that, unlike previous cohorts of older people, they are not financially dependent on their children, who have, in turn, benefitted from greater education and employment opportunities. Both G1 and G2 are now able to focus on promoting the life chances of G4. They believe that education is a method of retaining class advantage and these values are drivers of *jiaoyufication*.

Eldercare is another driver that cannot be ignored, and filial piety is strongly rooted in East Asian societies. However, the performance of filial obligations can be performed differently. The traditional forms of co-residence with the oldest generation can be replaced by "living-nearby-but-separate", or taking turns to live with the older parent, thus sharing care and allowing each daughter to feel that they are doing their loving duty. Meng et al. [40] found that those older people with more surviving children were less likely to be ageing in an institution. This study reveals that most families shared the common value of filial piety but negotiated different ways of performing this based on resources, such as family numbers, wealth, location, and time. How family care can be supported needs further investigation.

Finally, new forms of filial piety may be emerging where unquestioned expectations of care cannot be assumed but are, rather, reliant on adequate performance. Hence, young Mrs. Kong (G3) stated her refusal to provide eldercare for her parents given their negligent care of her as a mother and their grandparent duties. This can be viewed as a shift to a more egalitarian parent–child relationship that emphasizes emotional bonds and the notion of transactions rather than the filial obligations [41] found in G2–G3 relationships.

In conclusion, it can be found from the study that housing, particularly multiple home ownership, provides resources for negotiating ways of "doing family". In this social sustainability study, the spread of housing wealth across generations is increasing the transmission to the youngest family members. However, when care resources are seen through this intergenerational equity lens, it is clear that G2 emerges as a true sandwich generation, tasked by the duty to fulfil eldercare, and fuelled by middle-class motivations to support the youngest generation, but with the uncertain expectation that their own care needs can be met by a single child whose life focus has shifted.

**Author Contributions:** L.W. is the writer of this manuscript and drafted this paper. R.G. shaped the paper initially through critical comments and is also now contributing to the theoretical base. All authors have read and agreed to the published version of the manuscript.

**Funding:** This research received no external funding.

**Institutional Review Board Statement:** This research, drawn on a large doctoral thesis, is low-risk and was approved by Newcastle University on 25 September 2017. The study was conducted according to the guidelines of the Declaration of Newcastle University, and was approved by the Institutional Review Board of Newcastle University.

**Informed Consent Statement:** Informed consent was obtained from all participants involved in the study.

**Conflicts of Interest:** The authors declare no conflict of interest.

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
