# Peer review of "The Role of Housing in Facilitating Middle-Class Family Practices in China: A Case Study of Tianjin"

_sustainability, doi:10.3390/su132313031_

Round 1

Reviewer 1 Report

#Comments to authors

The manuscript, entitled “the role of housing in facilitating middle-class family practices in China: a case study of Tianjin,” aimed to examine the role of housing in facilitating traditional and emerging family care practices for older people and young child rearers. This is an interesting study corresponding to the problems of many countries, particularly developing countries where traditional practices, i.e., filial obligations, are similar to China’s.

  • Introduction
    • The authors provided clear information about the role of housing in China, particularly for aging people, lifestyle, and education. This is very interesting. However, the authors did not go deeply and specifically into the role of housing in facilitating middle-class family practices. We encourage you to review some literature on this matter.
  • Methods:
  • It was well structured.
  • It was hard to readers to understand how the authors classified the respondents as middle-class families because the authors did not provide information about the family’s income or other relevant indicators. Thus, it is important to provide some information about how the Chinese government classifies people into different social-economic classes.
  • It is important to explain why the authors chose Tianjin. Do people in this city practice differently from people in other cities?
  • Results and Conclusion
    • It was clear how the authors categorized into different themes. However, how can these findings be generalized? Does housing policy or the role of housing in facilitating people’s decision to purchase a house/flat differently throughout China?
  • Reference
    • Some references did not have a page number and publishing place. E.g., Murie, A. (2016) The Right to Buy? Selling off public and social housing.

Author Response

Dear Reviewer,

I am very glad to receive your notice about our manuscript and thanks very much to give us nice suggestions. We have made the correction of our manuscript and done the reply to the critique on a point-by-point basis. Please find the word file below.

Reviewer 2 Report

The article advances to an interesting field, which is, locate between two major topics of social coexistence in present times: This is the question for affordable housing within a capitalist market environment on the one hand and the increasing issues of eldercare due to the demographic change on the other hand. It uses a comprehensible and accessible language to describe its concerns and findings. It has a relatively broad theoretical foundation to introduce into the field of research. However, the theoretical width should be deepened in several points. In both above-mentioned fields – housing and demographic change – many extensive and well-founded researches already existing that could give the article a more robust anchoring within current debates.

By this step, also the existing structure - which makes sense in its mere function to subdivide the article - could get more sharpness of content. At some places and for some statements it would be appropriate to give more sources. For example at page 4: “Households are increasingly pursuing residential property as an investment strategy, as seen by multiple properties’ owners, international property investment, and buy-to-let landlordism, rather than a permanent shelter primarily for the family.“ Statements like this seem to be general knowledge in the recent debate. In order to approach them nevertheless in a scientifically critical way, the deeper theoretical embedding is recommended.

In order to give the empirical findings, which seem very elaborate and detailed in their collection, a stronger expression, it is recommended to illustrate the findings chapter, with more relevant information or, for example, by an overview graphic. By doing so, it could be easier for the authors to find more interfaces to other research programmes out of this field for the discussion chapter. Here only two sources are mentioned and a wider discussion is desirable.

Author Response

I am very glad to receive your notice about our manuscript and thank the reviewers to give us a nice suggestion. We have made the correction of our manuscript and done the reply to the critique on a point-by-point basis. Please see the word file below.

Reviewer 3 Report

The topic is relevant not only to China's situation, but also in many countries. This topic is really interesting. But still some suggestions for improvement:

  1. At the beginning (Abstract) it is recommended to add the research object, to make more clear research problem/question. 
  2. How this research is related to the context of sustainability approach? What indicators or research results reveal this?
  3. Why middle-class family practices are important - are they specific in China? How these families were selected (criteria???) for the research?
  4. What were the questions in the interview - we do not see it in the mthodology part. Then it is not clear the structure of results part.
  5. It is recommended to give logical scheme of the research-whether to see the steps of the research.
  6. The results could be more structured and conclusions could be more concrete and explore some sustainability insights.

Author Response

Dear reviewer,

I am very glad to receive your report about out manuscript and thanks to give us nice suggestions. We have made the correction of our manuscript and done the reply to the critique on a point-by-point basis. Please see the attachment below.

Reviewer 4 Report

The relevance to the journal title is not apparent. 
does not state the aim and objectives of the research 

theoretical contribution is lacking. The literature seems to explain trends but does not relate to theoretical concepts 

some English language grammatical errors 

Tables and Figures require description and discussion in the context of the study 

not clear how this is a case study  the method section requires more detail 

the aim and success of the research are unclear 

Author Response

Dear reviewer,

I am very glad to receive your report about our manuscript and thanks to give us nice suggestions. We have made the correction of our manuscript and done the reply to the critique on a point-by-point basis. Please see the word file below.

Reviewer 5 Report

The article deals with an interesting topic of welfare policies. I suggest to better explain the tool of the interview used, to better describe some constructs (such as "community") and to compare the situation of the families participating giving general information. Furthermore, a clear reference to the role of quality education in family life choices and in urban arrangements seems necessary.

Author Response

(The authors gave the same response as above.)

Round 2

Reviewer 3 Report

-

Reviewer 4 Report

Much improved, addresses the issues raised

Reviewer 5 Report

Thank you for the additions.